# Characteristics of Dietary Intake in Relation to the Consumption of Home-Produced Foods among Farm Women in Two Rural Areas of Kenya: A Preliminary Study

**Madoka Kishino** [1,2,3] **, Miru Hirose** [4] **, Azumi Hida** [5,]***, Yuki Tada** [5] **, Kazuko Ishikawa-Takata** [5] **, Kenta Hara** [6] **, Kenji Irie** [4] **, Patrick Maundu** [3] **and Yasuyuki Morimoto** [7]

1 Department of Food and Nutritional Science, Graduate School of Applied Bioscience, Tokyo University of Agriculture, 1-1-1 Sakuragaoka, Setagaya, Tokyo 156-8502, Japan
2 Research Fellow of Japan Society for the Promotion of Science, 5-3-1 Kojimachi, Chiyoda, Tokyo 102-0083, Japan
3 Kenya Resource Centre for Indigenous Knowledge, National Museums of Kenya, Kipande Road, Nairobi P.O. Box 40658-00100, Kenya
4 Department of International Agricultural Development, Faculty of International Agriculture and Food Studies, Tokyo University of Agriculture, 1-1-1 Sakuragaoka, Setagaya, Tokyo 156-8502, Japan
5 Department of Nutritional Science, Faculty of Applied Bioscience, Tokyo University of Agriculture, 1-1-1 Sakuragaoka, Setagaya, Tokyo 156-8502, Japan
6 Department of International Agricultural Development, Graduate School of International Food and Agricultural Studies, Tokyo University of Agriculture, 1-1-1 Sakuragaoka, Setagaya, Tokyo 156-8502, Japan
7 Diet Diversity for Nutrition and Health Program, Alliance of Bioversity International and International Center for Tropical Agriculture—CIAT, Nairobi P.O. Box 823-00621, Kenya
* Correspondence: a3hida@nodai.ac.jp; Tel.: +81-(0)3-5477-2378

**Abstract:** The present study aimed to clarify the differences in nutritional intake in relation to the consumption of local food products and dietary patterns between two rural Kenyan regions, Kitui and Vihiga, where different ethnic groups live in different agro-ecological zones. A participant observation study with weighted dietary records was conducted in August 2019. Enumerators stayed in each targeted household for approximately one week and measured the ingredients and dishes. We compared the dietary intake of farm women in charge of meal preparation ($n$ = 21) between the two regions and examined the contribution of each dish to the intake and the degree of home production for each food item. The results showed no difference in energy intake, but vitamin $B_2$, $B_{12}$, and C intakes were significantly higher in Vihiga, influenced by their consuming small fish and a variety of homegrown leafy vegetables. The people in Kitui consumed large quantities of homegrown pigeon peas, largely contributing to their nutritional intake. Dietary patterns were similar; common staple foods and tea with sugar accounted for about 40% of energy and protein intakes and fruit consumption was low. There was no difference in foods purchased frequently. These results suggested that promoting locally available fruits and vegetables would contribute to a sustainable supply of adequate micronutrients. Further studies are required to develop strategies to promote healthy dietary habits and improve health status.

**Keywords:** dietary habit; farmers; participant observation; food security; rural Kenya

## 1. Introduction

In Kenya, the risk of death and disability due to high blood pressure, body mass index (BMI), and blood sugar levels increased considerably from 2009 to 2019 [1]. Meanwhile, undernutrition, such as child stunting and anemia, is still an urgent problem [2,3]. A poor diet is a major contributing factor to the rising prevalence of malnutrition. In addition, being a multiethnic country with vast regional differences in agro-ecology, regionally specific dietary patterns may be associated with malnutrition. A previous study showed

that the prevalence of undernutrition and dietary patterns differed considerably among three Kenyan ethnic groups [4].

In 2019, the Food and Agriculture Organization of the United Nations and the World Health Organization recommended sustainable healthy diets, which are culturally acceptable dietary patterns that promote all dimensions of individuals' health and wellbeing, have a low environmental impact, and are accessible, affordable, safe, and equitable [5]. Considering local resources and dietary patterns is an essential aspect of nutritional intervention because it is culturally acceptable and helps maintain agricultural diversity. There are rich local crops in Africa that are suited to the country's natural features [6,7]. For example, African leafy vegetables are rich in vitamins and minerals and have the potential to contribute to nutritional improvement [8,9]. However, these are neglected in developed countries because of the different customs during colonization. Hence, there is little development in seed breeding and application technologies to meet the needs of local people. There are also limited studies evaluating the habitual intake of local food products in detail [6]. Keding et al. suggested that the diet pattern characterized by purchased foods was associated with higher BMI and the traditional dietary pattern was the healthiest in rural Tanzanian women [10]. Understanding how dietary patterns and food accessibility influence dietary intake is essential.

Therefore, understanding how local resources are utilized and contribute to an individual's nutritional intake and regionally specific dietary patterns may be essential to achieving sustainable healthy diets. This study aimed to examine the differences in nutritional intake between two rural Kenyan regions, Vihiga and Kitui, where different ethnic groups live in different agro-ecological zones. In addition, we examined the contribution of each dish to the nutrient intake and the degree of home production of each ingredient.

## 2. Materials and Methods

### 2.1. Study Design

This study was conducted as a basic survey of the Agrobiodiversity and Diet Diagnosis for Interventions Toolkit (ADD-IT) project [11], collaborating with Bioversity International and Tokyo University of Agriculture for nutrition improvement through the promotion of local agricultural resources. The survey period was seven days in Kitui and six days in Vihiga in August 2019, the dry season. A previous study showed no significant seasonal differences in food procurement and the degree of home consumption among smallholder farmers in Vihiga [12]. Body measurements were obtained at some point during the study, and dietary records were obtained through observation by enumerators. Twelve enumerators stayed in each targeted household during the survey period. The enumerators consisted of eight Japanese students from a university or graduate school (majoring in nutritional science or agricultural development), Kenyan staff from Bioversity International, and three Japanese Overseas Cooperation Volunteers. They received two days of training for the dietary survey in advance. The study was approved by the Ethics Committee of the Tokyo University of Agriculture (Approval No. 1818) and the KNH-UoN Ethics and Research Committee (Approval No. KNH-ERC/A/129). The study was conducted according to the code of ethics of the World Medical Association (Declaration of Helsinki).

### 2.2. Study Sites

The two project sites were selected based on their different agro-ecological zones. One was Kitui County, located in Eastern Kenya, approximately 160 km from Nairobi [13]. The county has a semi-arid climate with irregular and unreliable rainfall which ranges from about 400 mm to 1050 mm for annual precipitation depending on the location. The temperatures ranged between 13 °C to 33 °C. The absolute poverty level was 48% compared to the national average of 36% in 2016. The population density in Kitui was 37 persons per km$^2$ in 2019, and mainly the Kamba people live there. Major food crops grown in the county include cereals such as maize, sorghum, and millet; pulses such as green grams, cowpeas, and pigeon peas; root crops such as cassava, sweet potatoes, and taro; fruits such

as mangoes, papaya, and watermelons; and vegetables such as tomatoes, kales, onions, and bullet chilies. A well-known traditional dish of the Kamba is *Muthokoi*, the main ingredients being maize without husks and a pulse, commonly beans, pigeon peas or cowpeas.

The other site is Vihiga County, near Lake Victoria in western Kenya [14]. The county has one of the highest population densities in Kenya at 1033 persons per km$^2$, and mainly the Luhya people live here. The county is in a tropical zone, with an average annual precipitation of 1900 mm. The temperatures ranged between 14 °C and 32 °C. The main crops in the county include maize, beans, cassava, sweet potatoes, banana, vegetables, finger millet, sorghum, tea and a number of horticultural crops. The consumption of a type of small fish called *omena* was common in Vihiga [15]. In 2014, the prevalence of overweight or obese women was 31% in Kitui and 26% in Vihiga (33% nationally) [2]. Meanwhile, the child stunting rate was 46% in Kitui and 24% in Vihiga (26% nationwide) [2].

### 2.3. Participants

Targeted farmers were verbally informed about the study and then gave their consent. Potential households were selected through the connections of Bioversity International, ultimately selecting 11 households in Kitui and 12 in Vihiga based on safety, accessibility, and acceptance. The target individual for the dietary survey was an adult woman who engaged in meal preparation in each household. Dietary data were obtained from 10 women in the Kitui region and 11 from the Vihiga region. Two participants were excluded from the nutrient and food analyses because their records were insufficient.

### 2.4. Body Measurements

Investigators (YM and AH) visited and measured their height and weight using a tape measure and a weight scale (Digital health meter HD-660, TANITA, Tokyo, Japan) to check the health status of the targeted women. BMI was calculated using the following formula: weight (kg) divided by square of height (m$^2$).

### 2.5. Dietary Survey

In all target households, each dish was served in individual portions before eating. Enumerators recorded the dietary intake by participant observation. They weighed the foods, dishes and beverages as reference values using a digital cooking scale (KD-320, TANITA, Tokyo, Japan) and recorded individual portions for the targeted women as a proportion of the reference value. For example, if the measured reference value of cooked rice was 150 g and the subject woman's consumption was recorded as "2 times as the reference", the consumed amount was estimated to be about 300 g. Raw ingredients, including water, condiments, and cooking oil for the whole family, leftovers, and inedible portions (peels, seeds, bone, among others) were measured and cooking methods were described as much as possible. When the targeted woman ate outside, the enumerators asked for ingredients and intake descriptions and then estimated. In addition, the time to eat and the source of each ingredient (own-produced, purchased, gift, or other) were recorded.

### 2.6. Dietary Evaluation

We calculated the daily energy and nutrient intake per person using the Kenya Food Composition Tables (KFCT) [16], using the food code of the cooked ingredients (boiled or stewed, etc.) according to the recorded cooking method as much as possible. Kenya has commercially available nutritional fortified products such as maize and wheat flour, salt, oil, and fat [17]. In this study, we carried out nutritional calculations with fortified wheat flour, iodized salt, fortified margarine, unfortified maize flour, and cooking oil. All participants purchased commercial wheat flour and margarine but did not purchase maize in many cases in this survey. The cooking oils had no labels, and we could not determine whether these products were fortified. According to a national survey, iodized salt consumption is 95% [2]. The percentage of individuals at risk of insufficient or excess nutrient intake was calculated based on whether an individual's average intake met the reference level for each

nutrient [18]. Days when at least one meal per day could not be estimated were excluded from the calculation. The dietary data period was 4.3 days for Kitui and 3.3 days for Vihiga on average.

To examine mealtimes, we calculated energy intake per time of day and pre-defined meal timings (6:00–10:00 for breakfast, 12:00–15:00 for lunch, 18:00–21:00 for dinner, and other times as snacks) [19]. Meal frequency was calculated by summing all eating and drinking behaviors.

All the ingredients were classified into 18 food groups. Groupings were based on KFCT [16], with vegetables further divided into green and yellow vegetables and other vegetables, and meat, poultry, and eggs further divided into meat and poultry, and eggs. Confectionaries (biscuits and cake) were added. The intake (g) of each food group was calculated.

In addition, we categorized all dishes based on ingredients, cooking methods, and common dish names. The frequency, median intake, and contribution rate to the total energy intake, protein, calcium, iron, vitamin A, vitamin $B_{12}$, and vitamin C for each dish were calculated. Iron was selected because it is a nutrient related to anemia [3]. Calcium and vitamins were chosen because of the high risk of insufficiency in this study. We listed dishes that included 90% of the cumulative contribution to energy and five nutrients.

Finally, the degree of home production of each food item was examined by the average frequency of home production and frequency of use (times per day). We listed commonly consumed items as more than 0.05 times per day, excluding tea leaves, coffee, sugar, seasonings, and cooking oils.

### 2.7. Statistical Analysis

Data are presented as mean $\pm$ standard deviation. The differences between regions were examined using the non-paired *t*-test or Mann–Whitney U test for continuous variables based on the results of the Shapiro–Wilk test (normality test) or Fisher's exact test for categorical variables. Statistical analyses were performed using the IBM SPSS Statistics ver. 28 (IBM Corp., Armonk, NY, USA).

### 3. Results

Table 1 shows the characteristics of the participants. There were no regional differences in age or BMI. The prevalence of obesity was high in both regions at 60% in Kitui and 36% in Vihiga.

**Table 1.** The characteristics of participants (farm women in charge of meal preparation).

| | Kitui (*n* = 10) | | Vihiga (*n* = 11) | | *p*-Value * |
|---|---|---|---|---|---|
| Age (years old) | 47.0 $\pm$ 12.6 | (30–70) | 44.5 $\pm$ 14.2 | (25–64) | 0.670 |
| BMI (kg/m$^2$) | 29.5 $\pm$ 4.3 | (19.4–35.8) | 27.7 $\pm$ 5.4 | (20.3–37.0) | 0.423 |
| Overweight (25 $\leq$ BMI < 30) | 3 (30%) | | 4 (36%) | | |
| Obese (30 $\leq$ BMI) | 6 (60%) | | 4 (36%) | | |

Values are expressed as mean $\pm$ SD (min–max) or number of participants (%). * non-paired *t*-test.

Table 2 presents the dietary intakes. There was no significant difference in daily energy intake, but Kitui had a significantly lower intake of vitamins $B_2$, $B_{12}$, and C and a higher risk of vitamin $B_{12}$ and C insufficiency. More than 50% of people did not meet the reference levels of calcium and vitamin A for both regions and vitamins $B_{12}$ and C in Kitui only. Table 3 shows the food group intake. In Kitui, beans and pulses intake was significantly higher and fish intake was lower than in Vihiga.

There was no significant difference in average meal frequency with 3.3 $\pm$ 0.4 for Kitui and 3.6 $\pm$ 1.2 (*p* = 0.159) for Vihiga. In Kitui, mealtime was between 6:00 and 22:00, and peaks were at 7:00–8:00, 10:00, 12:00–14:00, and 19:00–21:00 (Figure 1). In Vihiga, mealtimes were between 7:00 and 22:00, and peaks were observed at 8:00, 13:00–14:00, and 19:00–20:00. During breakfast (6:00–10:00), energy intake tended to be lower in Kitui (Table 2).

**Table 2.** Daily energy and nutrient intakes in farm women in charge of meal preparation.

| | Kitui (*n* = 10) | | Vihiga (*n* = 11) | | *p*-Value for Intake [1] | *p*-Value for Risk [2] |
|---|---|---|---|---|---|---|
| | Mean ± SD | Risk of Insufficient or Excess | Mean ± SD | Risk of Insufficient or Excess | | |
| Total energy (kcal) [†] | 1922 ± 421 | - | 2123 ± 598 | - | 0.573 | - |
| Breakfast time (kcal) [3] | 444 ± 195 | - | 603 ± 193 | - | 0.075 | - |
| Lunch time (kcal) [3] | 619 ± 191 | - | 612 ± 291 | - | 0.951 | - |
| Dinner time (kcal) [3] | 716 ± 165 | - | 615 ± 257 | - | 0.303 | - |
| Snack (kcal) [3][†] | 143 ± 164 | - | 292 ± 402 | - | 0.245 | - |
| Protein (g) | 61.1 ± 20.2 | - | 60.7 ± 16.6 | - | 0.956 | - |
| Protein (g/kgBW) [†] | 0.82 ± 0.3 | 20.0% | 0.88 ± 0.3 | 18.2% | 0.324 | 1.000 |
| Fat (g) [†] | 50.0 ± 20.8 | - | 59.6 ± 23.7 | - | 0.336 | - |
| Carbohydrate (g) | 284.8 ± 54.2 | - | 317.8 ± 88.4 | - | 0.313 | - |
| Protein/energy ratio (%) | 12.7 ± 2.2 | 10.0% | 11.5 ± 1.1 | 9.1% | 0.163 | 1.000 |
| Fat/energy ratio (%) | 22.4 ± 6.4 | 40.0% | 24.9 ± 4.3 | 9.1% | 0.332 | 0.149 |
| Carb/energy ratio (%) | 64.9 ± 6.0 | 40.0% * | 63.6 ± 4.7 | 45.5% * | 0.607 | 1.000 |
| Dietary fiber (g) [†] | 43.4 ± 15.4 | 20.0% | 35.5 ± 8.0 | 27.3% | 0.149 | 1.000 |
| Calcium (mg) [†] | 608 ± 248 | 90.0% | 884 ± 412 | 81.8% | 0.121 | 1.000 |
| Magnesium (mg) | 292 ± 94 | 30.0% | 315 ± 59 | 27.3% | 0.509 | 1.000 |
| Iron (mg) | 16.8 ± 4.4 | 30.0% | 20.0 ± 5.2 | 18.2% | 0.149 | 0.635 |
| Zinc (mg) | 8.7 ± 2.8 | 20.0% | 9.8 ± 2.4 | 9.1% | 0.361 | 0.586 |
| Vitamin A (μgRAE) | 257 ± 109 | 100.0% | 379 ± 159 | 81.8% | 0.056 | 0.476 |
| Vitamin $B_1$ (mg) | 1.51 ± 0.43 | 10.0% | 1.41 ± 0.52 | 18.2% | 0.652 | 1.000 |
| Vitamin $B_2$ (mg) [†] | 1.21 ± 0.47 | 20.0% | 1.93 ± 1.02 | 9.1% | 0.020 | 0.586 |
| Niacin (mg) | 13.1 ± 4.0 | 30.0% | 14.8 ± 4.4 | 9.1% | 0.354 | 0.311 |
| Vitamin $B_{12}$ (μg) [†] | 2.2 ± 1.2 | 60.0% | 4.9 ± 3.7 | 9.1% | 0.017 | 0.024 |
| Folate (μg) | 625 ± 231 | 10.0% | 690 ± 354 | 9.1% | 0.625 | 1.000 |
| Vitamin C (mg) | 48 ± 26 | 80.0% | 79 ± 31 | 27.3% | 0.022 | 0.030 |

BW: body weight, Carb: carbohydrate, RAE: retinol activity equivalents, Values are expressed as mean ± SD or percentage of those who did not meet acceptable macronutrient distribution ranges (AMDR), estimated average requirement, or adequate intake (* Those above the AMDR.) [15]. The iron requirement was adjusted from 18% bioavailability to 10% by multiplying it by 1.8. [1] non-paired *t*-test or [†] Mann–Whitney U test, [2] Fisher's exact test, [3] breakfast, lunch, dinner, and snacks were defined as eating between 6:00–10:00, 12:00–15:00, 18:00–21:00 and other timings, respectively.

**Table 3.** Daily food group intakes in farm women in charge of meal preparation.

| | Kitui (*n* = 10) | Vihiga (*n* = 11) | *p*-Value * |
|---|---|---|---|
| | Mean ± SD | Mean ± SD | |
| Grains and cereals | 541.3 ± 147.8 | 633.9 ± 209.3 | 0.260 |
| Starchy roots, bananas and tubers [†] | 50.0 ± 34.8 | 112.6 ± 101.2 | 0.397 |
| Sugar and sweeteners | 27.4 ± 16.9 | 51.3 ± 49.4 | 0.155 |
| Nuts and seeds [†] | 2.2 ± 6.3 | 5.4 ± 9.3 | 0.208 |
| Green and yellow vegetables [1] | 34.8 ± 24.8 | 52.8 ± 36.6 | 0.209 |
| Other vegetables | 91.9 ± 38.1 | 111.5 ± 57.5 | 0.374 |
| Fruits [†] | 19.6 ± 26.1 | 27.4 ± 25.4 | 0.420 |
| Beans and pulses | 220.2 ± 160.3 | 49.5 ± 50.2 | 0.008 |
| Fishes [†] | 2.0 ± 6.3 | 13.0 ± 15.1 | 0.004 |
| Meats and poultries | 28.7 ± 25.4 | 25.7 ± 23.5 | 0.780 |
| Eggs [†] | 5.3 ± 7.7 | 8.3 ± 10.0 | 0.447 |
| Milk and dairy products [†] | 120.0 ± 75.8 | 287.6 ± 293.7 | 0.121 |
| Oils and fats | 25.5 ± 14.8 | 23.4 ± 13.8 | 0.751 |
| Confectioneries [†] | 0.4 ± 1.1 | 6.9 ± 13.4 | 0.259 |
| Beverages [†] | 412.9 ± 242.4 | 608.4 ± 387.1 | 0.205 |
| Condiments | 9.9 ± 5.4 | 7.1 ± 2.6 | 0.171 |

Unit: grams per day, Values are expressed as mean ± SD. * non-paired *t*-test, [†] Mann–Whitney U test. [1] Green and yellow vegetables were defined based on nutrient contents (more than 600 μg of carotene per 100 g edible portion).

Table 4 shows the frequency of consumption and the median intake for each dish. Twenty-four dishes were listed for Kitui. They often consumed bread or *mandazi* (doughnut) with milk tea at breakfast, *muthokoi*, *ugali* (Kenyan staple food made from maize flour kneaded with hot water), *chapati*, rice, pigeon pea stew, or stir-fried kales at lunch and dinner. For Vihiga, 35 dishes were used. They often consumed *mandazi*, bread, maize, or cassava with milk tea at breakfast, *ugali*, rice, beef stew, *omena* (small fish) stew, fried egg, stir-fried kales or cabbage, and milk tea at lunch and dinner. In both regions, they drank milk tea at all times but slightly more frequently in Vihiga, especially during snacking.

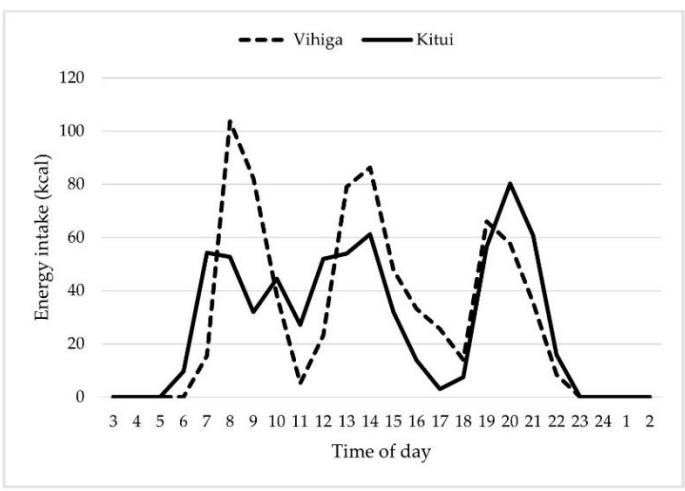

**Figure 1.** Average energy intake per time of day in farm women in charge of meal preparation (Kitui: 10 women, Vihiga: 11 women).

**Table 4.** Frequency and median intake of each dish/food in farm women in charge of meal preparation.

| Group | Name of Dishes/Foods | Main Ingredients [1] | Kitui | | | | | | Vihiga | | | | | |
|---|---|---|---|---|---|---|---|---|---|---|---|---|---|---|
| | | | Frequency (Times/Day) [3] | | | | | Median [4] (g/Time) | Frequency (Times/Day) [3] | | | | | Median [4] (g/Time) |
| | | | Total | BF | LC | DN | SN | | Total | BF | LC | DN | SN | |
| Staple foods | *Ugali* | Maize flour | 0.64 | 0.02 | 0.19 | 0.36 | 0.06 | 325 | 1.10 | 0.03 | 0.28 | 0.63 | 0.18 | 300 |
| | *Muthokoi* | Degermed maize and beans | 0.45 | 0.06 | 0.13 | 0.26 | 0.00 | 490 | - | - | - | - | - | - |
| | *Mandazi* (doughnut) | (mostly commercial) | 0.40 | 0.28 | 0.09 | 0.02 | 0.02 | 100 | 0.48 | 0.40 | 0.05 | 0.00 | 0.03 | 102 |
| | Boiled rice | - | 0.30 | 0.02 | 0.15 | 0.11 | 0.02 | 300 | 0.35 | 0.03 | 0.25 | 0.08 | 0.00 | 336 |
| | *Chapati* | Wheat flour (and sugar) | 0.32 | 0.09 | 0.11 | 0.11 | 0.02 | 195 | 0.30 | 0.10 | 0.08 | 0.08 | 0.05 | 137 |
| | Bread | (mostly commercial) | 0.45 | 0.43 | 0.00 | 0.00 | 0.02 | 58 | 0.28 | 0.15 | 0.03 | 0.03 | 0.08 | 82 |
| | Rice and beans | Rice and beans | 0.23 | 0.04 | 0.09 | 0.09 | 0.02 | 434 | - | - | - | - | - | - |
| | Boiled or roasted maize | - | - | - | - | - | - | - | 0.20 | 0.13 | 0.05 | 0.03 | 0.00 | 200 |
| | *Githeri* | Maize and beans | 0.06 | 0.00 | 0.02 | 0.04 | 0.00 | 300 | 0.15 | 0.08 | 0.05 | 0.03 | 0.00 | 392 |
| | Cassava | - | - | - | - | - | - | - | 0.13 | 0.10 | 0.03 | 0.00 | 0.00 | 110 |
| | Rice and potato | Rice, Irish potato, T&O | 0.09 | 0.02 | 0.04 | 0.02 | 0.00 | 313 | 0.05 | 0.03 | 0.00 | 0.03 | 0.00 | 325 |
| | *Matoke* (green banana) | Green banana, T&O | 0.02 | 0.00 | 0.02 | 0.00 | 0.00 | 357 | 0.10 | 0.03 | 0.03 | 0.00 | 0.05 | 550 |
| | Potato and cabbage stew | Irish potato, cabbage, T&O | 0.06 | 0.00 | 0.04 | 0.02 | 0.00 | 150 | - | - | - | - | - | - |
| | Sweet potato | - | - | - | - | - | - | - | 0.05 | 0.00 | 0.05 | 0.00 | 0.00 | 225 |
| | Spaghetti | - | - | - | - | - | - | - | 0.05 | 0.00 | 0.03 | 0.03 | 0.00 | 330 |
| | Banana and beans | Green banana and beans | 0.04 | 0.00 | 0.04 | 0.00 | 0.00 | 536 | - | - | - | - | - | - |
| | Potato and banana | Irish potato and green banana | - | - | - | - | - | - | 0.03 | 0.00 | 0.03 | 0.00 | 0.00 | 500 |
| ASFs | Beef stew | Beef meat, T&O | 0.11 | 0.00 | 0.02 | 0.06 | 0.02 | 250 | 0.38 | 0.00 | 0.18 | 0.18 | 0.03 | 63 |
| | Goat stew | Goat meat, T&O | 0.17 | 0.00 | 0.06 | 0.06 | 0.04 | 150 | - | - | - | - | - | - |
| | *Omena* (small fishes) stew | *Omena*, T&O | - | - | - | - | - | - | 0.20 | 0.00 | 0.03 | 0.13 | 0.05 | 55 |
| | Fried egg | Chicken egg, T&O | - | - | - | - | - | - | 0.18 | 0.05 | 0.00 | 0.10 | 0.03 | 50 |
| | Boiled Egg | Chicken egg | 0.09 | 0.09 | 0.00 | 0.00 | 0.00 | 51 | - | - | - | - | - | - |
| | Fish, samaki, tilapia | Fish, T&O | - | - | - | - | - | - | 0.10 | 0.00 | 0.00 | 0.08 | 0.03 | 174 |
| | Chicken stew | Chicken meat, T&O | 0.09 | 0.00 | 0.00 | 0.09 | 0.00 | 235 | 0.05 | 0.00 | 0.03 | 0.00 | 0.03 | 200 |
| Beans and pulses | Pigeon pea stew | Pigeon pea, T&O | 0.32 | 0.02 | 0.13 | 0.13 | 0.04 | 220 | - | - | - | - | - | - |
| | Green gram stew | Green grams, T&O | - | - | - | - | - | - | 0.13 | 0.00 | 0.03 | 0.10 | 0.00 | 257 |
| Stir-fried vegetables | Kales (*Sukuma wiki*) | Kales, T&O | 0.36 | 0.00 | 0.13 | 0.23 | 0.00 | 150 | 0.40 | 0.00 | 0.13 | 0.28 | 0.00 | 110 |
| | Cabbage | Cabbage, T&O | 0.15 | 0.00 | 0.09 | 0.06 | 0.00 | 125 | 0.30 | 0.00 | 0.18 | 0.08 | 0.05 | 143 |
| | Amaranth vegetables | Amaranth leaves, T&O | - | - | - | - | - | - | 0.18 | 0.00 | 0.08 | 0.08 | 0.03 | 66 |
| | Mixed leaves | Green leaves [2], T&O | - | - | - | - | - | - | 0.15 | 0.00 | 0.03 | 0.08 | 0.05 | 128 |
| | Cowpea leaves | Cowpea leaves, T&O | - | - | - | - | - | - | 0.03 | 0.00 | 0.00 | 0.03 | 0.00 | 287 |
| | Carrot | - | - | - | - | - | - | - | 0.03 | 0.00 | 0.03 | 0.00 | 0.00 | 43 |
| Fruits | Avocado | - | 0.13 | 0.00 | 0.09 | 0.04 | 0.00 | 75 | 0.05 | 0.03 | 0.00 | 0.03 | 0.00 | 175 |
| | Papaya | - | 0.09 | 0.00 | 0.06 | 0.02 | 0.00 | 175 | - | - | - | - | - | - |
| | Mango | - | - | - | - | - | - | - | 0.08 | 0.03 | 0.00 | 0.00 | 0.05 | 105 |
| | Orange | - | - | - | - | - | - | - | 0.08 | 0.00 | 0.03 | 0.03 | 0.03 | 90 |
| | Guava | - | - | - | - | - | - | - | 0.03 | 0.00 | 0.00 | 0.00 | 0.03 | 120 |
| Nuts | Groundnut | - | - | - | - | - | - | - | 0.20 | 0.08 | 0.05 | 0.03 | 0.05 | 22 |
| Beverages | Milk tea | Tea and milk (cow or goat) | 1.47 | 0.91 | 0.17 | 0.21 | 0.17 | 400 | 1.70 | 0.90 | 0.30 | 0.13 | 0.38 | 400 |
| | Miro | Milk and miro powder | - | - | - | - | - | - | 0.05 | 0.05 | 0.00 | 0.00 | 0.00 | 300 |
| Others | Sugar in milk tea | - | 1.47 | 0.91 | 0.17 | 0.21 | 0.17 | 10 | 1.55 | 0.75 | 0.33 | 0.10 | 0.38 | 18 |
| | Margarine on the bread | - | 0.19 | 0.17 | 0.00 | 0.00 | 0.02 | 9 | 0.23 | 0.13 | 0.03 | 0.03 | 0.05 | 8 |
| | Sugar in black tea | - | - | - | - | - | - | - | 0.15 | 0.05 | 0.03 | 0.05 | 0.03 | 65 |

BF: Breakfast, LC: Lunch, DN: Dinner, SN: Snack, ASFs: Animal-source foods, T&O: Tomato and onion. [1] excluding salt and oil, [2] cowpeas leaves (*kunde*), black nightshade (*managu*), pumpkin leaves (*seveve*), amaranth leaves (*mchicha*), vine spinach (*nderema*), jute mallow (*mrenda*) and slender leaves (*mitoo*), [3] Breakfast, Lunch and Dinner were defined as eating between 6:00–10:00, 12:00–15:00 and 18:00–21:00, respectively. Snack was defined as eating during other timing, [4] Median amount of serving.

Table 5 shows the contribution rates to the total energy and nutrient intake for each dish. For Kitui, *muthokoi*, *ugali*, and *chapati* contributed more energy, protein, calcium, iron, and other micronutrients. Milk tea with sugar had high contributions with 10% for energy, 8% for protein, 28% for calcium, 22% for vitamin A and 54% for vitamin $B_{12}$. Stir-fried kale (38%) and papaya (10%) significantly contributed to vitamin C intake. For Vihiga, *ugali* had the highest contributions to energy, protein, iron, and zinc, at more than 20%. Milk tea with sugar was also high, with 17% energy, 18% protein, 44% calcium, 22% zinc, 34% vitamin A, and 58% vitamin $B_{12}$. Stir-fried vegetables were essential sources of calcium (26%), vitamin A (28%), and vitamin C (46%). Fruit is also an essential source of vitamin C (14%).

**Table 5.** Contribution rate to total nutrient intake of each dish/food in farm women in charge of meal preparation.

| Group | Name of Dish/Food | Kitui | | | | | | | Vihiga | | | | | | |
|---|---|---|---|---|---|---|---|---|---|---|---|---|---|---|---|
| | | En | Pro | Ca | Iron | V.A | V.B$_{12}$ | V.C | En | Pro | Ca | Iron | V.A | V.B$_{12}$ | V.C |
| Staple foods | *Ugali* | 15.4 | 11.2 | 4.0 | 13.7 | 0.0 | 0.0 | 0.0 | 24.5 | 20.1 | 5.1 | 20.2 | 0.1 | 0.0 | 0.1 |
| | *Muthokoi* | 16.0 | 24.1 | 14.8 | 17.7 | 5.1 | 0.0 | 3.3 | - | - | - | - | - | - | - |
| | *Mandazi* (doughnut) | 7.7 | 4.4 | 9.2 | 8.8 | 7.7 | 7.4 | 0.0 | 7.5 | 5.0 | 7.2 | 8.2 | 5.9 | 4.0 | 0.0 |
| | Boiled rice | 5.9 | 3.4 | 0.7 | 3.1 | 0.0 | 0.0 | 0.0 | 7.2 | 4.4 | 0.7 | 3.5 | 1.4 | 0.0 | 1.2 |
| | *Chapati* [1] | 11.7 | 6.8 | 2.2 | 12.4 | 18.4 | 11.5 | 0.0 | 8.4 | 7.6 | 1.6 | 11.5 | 10.2 | 6.1 | 0.0 |
| | Bread | 3.8 | 3.7 | 1.7 | 3.0 | 0.0 | 0.9 | 0.0 | 2.8 | 3.0 | 1.0 | 2.0 | 0.0 | 0.4 | 0.0 |
| | Rice and beans | 5.8 | 5.8 | 0.3 | 5.1 | 0.2 | 0.0 | 0.9 | - | - | - | - | - | - | - |
| | Maize | - | - | - | - | - | - | - | 0.9 | 0.8 | 0.2 | 0.7 | 0.0 | 0.0 | 0.0 |
| | *Githeri* | 1.8 | 2.1 | 1.2 | 2.0 | 0.7 | 0.0 | 0.6 | 2.6 | 4.2 | 1.3 | 3.5 | 0.4 | 0.0 | 0.4 |
| | Cassava | - | - | - | - | - | - | - | 1.2 | 0.3 | 0.5 | 0.6 | 0.0 | 0.0 | 4.6 |
| | Rice and potato | 1.8 | 1.2 | 0.0 | 1.4 | 0.0 | 0.0 | 1.6 | 0.9 | 0.6 | 0.1 | 0.8 | 0.0 | 0.0 | 0.6 |
| | *Matoke* | 0.3 | 0.1 | 0.1 | 0.2 | 0.0 | 0.0 | 2.2 | 2.6 | 1.4 | 0.5 | 2.3 | 0.1 | 0.0 | 13.7 |
| | Potato and cabbage stew | 0.5 | 0.8 | 1.3 | 0.6 | 0.3 | 1.1 | 2.0 | - | - | - | - | - | - | - |
| | Sweet potato | - | - | - | - | - | - | - | 0.5 | 0.3 | 0.3 | 0.2 | 0.2 | 0.0 | 1.7 |
| | Spaghetti | - | - | - | - | - | - | - | 0.9 | 1.0 | 0.1 | 0.2 | 0.0 | 0.0 | 0.0 |
| | Banana and beans | 1.3 | 1.8 | 3.1 | 1.7 | 1.2 | 0.0 | 3.9 | - | - | - | - | - | - | - |
| | Potato and banana | - | - | - | - | - | - | - | 0.6 | 0.4 | 0.1 | 0.6 | 0.0 | 0.0 | 1.9 |
| ASFs | Beef stew | 1.3 | 3.2 | 0.3 | 6.3 | 0.9 | 5.1 | 2.4 | 2.4 | 8.7 | 0.2 | 16.6 | 0.9 | 7.2 | 1.5 |
| | Goat stew | 1.6 | 4.5 | 0.4 | 2.1 | 1.4 | 8.1 | 3.3 | - | - | - | - | - | - | - |
| | *Omena* (small fishes) stew | - | - | - | - | - | - | - | 0.6 | 1.3 | 4.1 | 0.6 | 0.7 | 14.3 | 0.9 |
| | Fried egg | - | - | - | - | - | - | - | 0.7 | 1.7 | 0.5 | 0.8 | 4.1 | 2.9 | 0.3 |
| | Boiled Egg | 0.3 | 0.8 | 0.3 | 0.4 | 2.8 | 1.8 | 0.0 | - | - | - | - | - | - | - |
| | Fish, samaki, tilapia | - | - | - | - | - | - | - | 1.2 | 3.1 | 1.2 | 1.4 | 0.7 | 2.9 | 0.9 |
| | Chicken stew | 1.3 | 2.4 | 0.3 | 1.5 | 2.5 | 2.0 | 2.0 | 0.9 | 2.0 | 0.1 | 0.9 | 1.4 | 0.9 | 0.5 |
| Beans and pulses | Pigeon pea stew | 4.3 | 8.0 | 8.4 | 7.3 | 3.0 | 0.0 | 6.2 | - | - | - | - | - | - | - |
| | Green gram stew | - | - | - | - | - | - | - | 1.9 | 4.3 | 1.5 | 3.2 | 1.0 | 0.0 | 1.2 |
| Stir-fried vegetables | Kales (*Sukuma wiki*) | 1.9 | 2.3 | 18.0 | 4.6 | 18.9 | 0.9 | 37.5 | 1.5 | 1.9 | 14.2 | 3.9 | 14.7 | 0.0 | 23.2 |
| | Cabbage | 0.5 | 0.6 | 1.0 | 0.6 | 0.3 | 0.7 | 6.1 | 1.1 | 0.8 | 1.9 | 1.0 | 0.9 | 0.0 | 9.7 |
| | Amaranth vegetables | - | - | - | - | - | - | - | 0.4 | 0.6 | 2.7 | 2.3 | 6.9 | 0.0 | 3.7 |
| | Mixed leaves | - | - | - | - | - | - | - | 0.8 | 1.3 | 5.3 | 3.7 | 4.9 | 0.0 | 7.6 |
| | Cowpea leaves | - | - | - | - | - | - | - | 0.2 | 0.3 | 1.2 | 0.6 | 0.0 | 0.0 | 1.4 |
| | Carrot | - | - | - | - | - | - | - | 0.0 | 0.0 | 0.0 | 0.0 | 1.1 | 0.0 | 0.0 |
| Fruits | Avocado | 1.0 | 0.3 | 0.3 | 0.6 | 0.1 | 0.0 | 3.0 | 0.7 | 0.2 | 0.2 | 0.4 | 0.0 | 0.0 | 1.4 |
| | Papaya | 0.2 | 0.1 | 0.3 | 0.4 | 3.2 | 0.0 | 10.4 | - | - | - | - | - | - | - |
| | Mango | - | - | - | - | - | - | - | 0.2 | 0.1 | 0.1 | 0.1 | 1.6 | 0.0 | 2.7 |
| | Orange | - | - | - | - | - | - | - | 0.1 | 0.0 | 0.1 | 0.0 | 0.0 | 0.0 | 1.6 |
| | Guava | - | - | - | - | - | - | - | 0.1 | 0.1 | 0.1 | 0.1 | 0.2 | 0.0 | 8.1 |
| Nuts | Groundnut | - | - | - | - | - | - | - | 1.6 | 1.9 | 0.8 | 1.6 | 0.0 | 0.0 | 0.0 |
| Beverages | Milk tea | 5.3 | 8.2 | 28.4 | 2.9 | 21.5 | 53.5 | 7.7 | 10.7 | 18.3 | 43.9 | 4.0 | 33.6 | 57.8 | 6.3 |
| | Miro | - | - | - | - | - | - | - | 0.3 | 0.5 | 1.2 | 0.0 | 0.9 | 1.6 | 0.0 |
| Others | Sugar in milk tea | 5.1 | 0.0 | 0.0 | 0.0 | 0.0 | 0.0 | 0.0 | 6.2 | 0.0 | 0.0 | 0.0 | 0.0 | 0.0 | 0.0 |
| | Margarine on the bread [1] | 0.6 | 0.0 | 0.0 | 0.0 | 7.0 | 2.5 | 0.0 | 0.5 | 0.0 | 0.0 | 0.0 | 5.1 | 1.2 | 0.0 |
| | Sugar in black tea | - | - | - | - | - | - | - | 2.2 | 0.0 | 0.0 | 0.0 | 0.0 | 0.0 | 0.0 |

En: Energy, Pro: Protein, Ca: Calcium, V.A: Vitamin A, V.B$_{12}$: Vitamin $B_{12}$, V.C: Vitamin C, ASFs: Animal-source foods, Value shows the contribution rate (%) to total intake for each nutrient. All nutritional values are based on Kenya Food Composition tables 2018, bold font indicates items included in the 90% contribution to each nutrient. [1] calculated with fortified products.

In total, 34 out of 66 items in 11 households in Kitui and 32 out of 64 items in 12 households in Vihiga were self-sufficient at least once. In both regions, milk, tomatoes, onions, and maize consumption was very high; however, production frequencies were not very high, except for maize (Figures 2 and 3). In Kitui (Figure 2), the consumption of pigeon peas was high and more than 90% was produced. In Vihiga (Figure 3), consumption of African leafy vegetables was relatively low. However, the frequency of home production was high, with 57% for cowpea leaves (*kunde*), 80% for slender leaves (*mitoo*), 75% for pumpkin leaves (*seveve*), and 67% for black nightshade (*managu*). On the other hand, although the frequencies of consumption of *mandazi*, rice, bread, cabbage, and Irish potato were high in both regions, the frequency of home production was close to zero (mostly purchased).

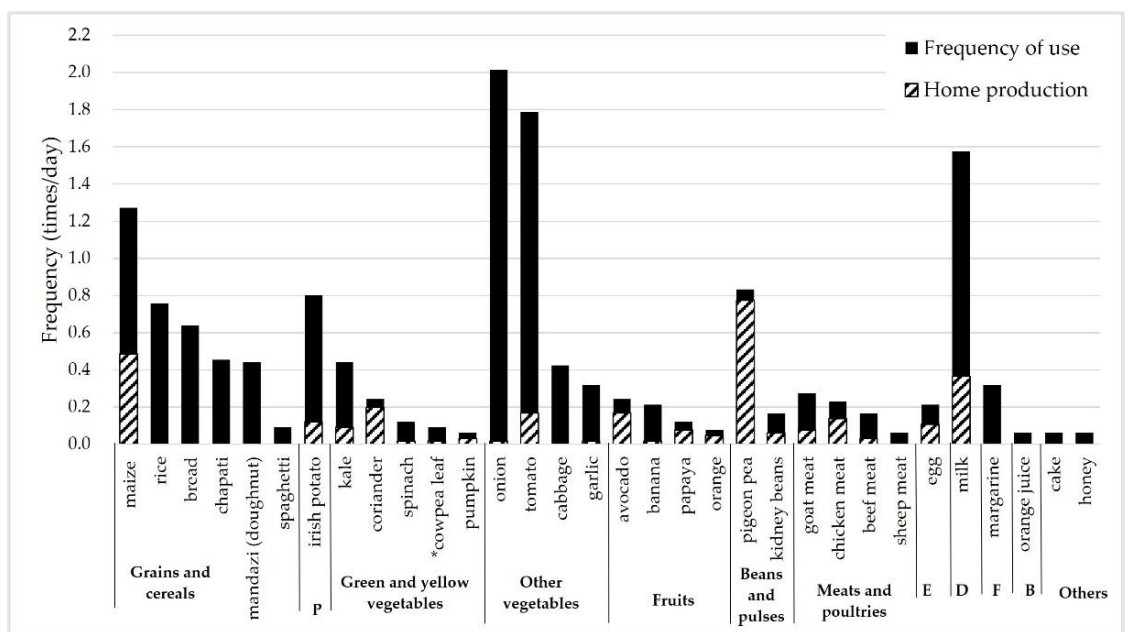

**Figure 2.** Frequency of consumption and home production of each food item in Kitui. (Bold font indicates food groups. P: Potatoes and starches, E: Eggs, D: Dairy products, F: Fats, B: Beverages. * African leafy vegetables, A total of 66 items were consumed, and 34 items were homegrown or produced in 11 households in Kitui. Food rarely consumed (less than 0.05 times per day), tea leaves, coffee, sugar, salt, *royco* (seasoning), and cooking oil were excluded.).

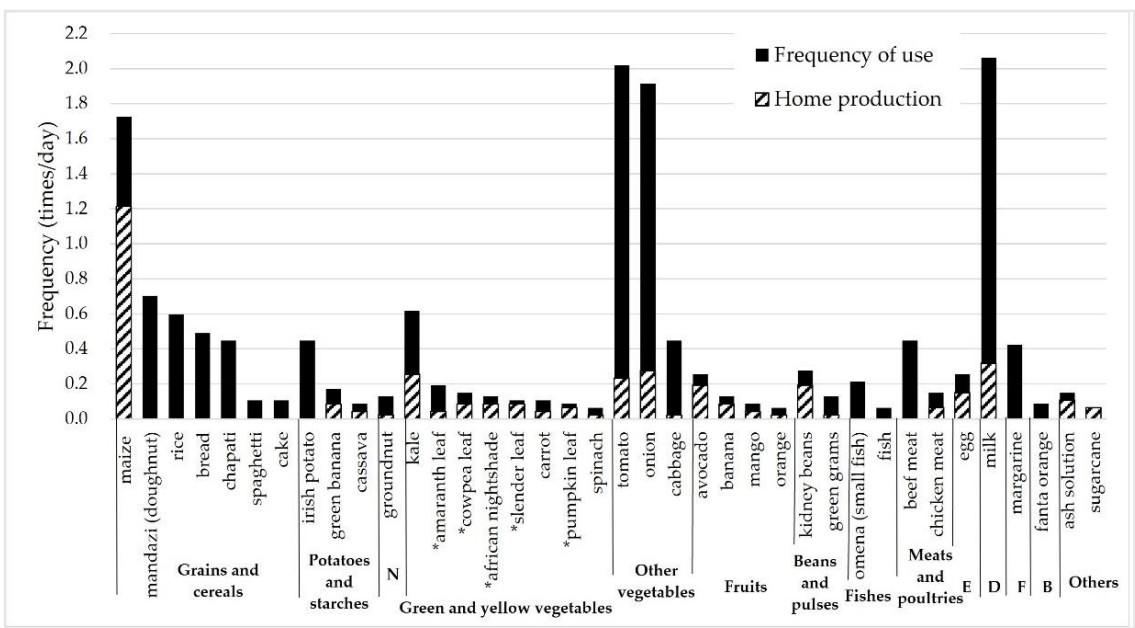

**Figure 3.** Frequency of consumption and home production of each food item in Vihiga. (Bold font indicates food groups. N: Nuts and seeds, E: Eggs, D: Dairy products, F: Fats, B: Beverages. * African leafy vegetables, A total of 64 items were consumed, 32 items were homegrown or produced in 12 households in Vihiga. Foods rarely consumed (less than 0.05 times per day), tea leaves, coffee, sugar, salt, *royco* (seasoning), and cooking oil were excluded.).

## 4. Discussion

The present study clarified the differences in nutritional intake and examined the influence of local products and dietary patterns on the intake of farm women responsible

for food preparation in the two regions. Many participants in both areas were overweight or obese and at moderate to high risk for insufficient micronutrient intake. There were no significant differences in mean energy intake but the participants in Vihiga had higher intakes of vitamins $B_2$, $B_{12}$, and C. That was due to regional-specific food intakes such as high fish consumption and a greater variety of fruits and African leafy vegetables in Vihiga. Food consumption was not necessarily related to their home production except for some regional-specific foods. Dietary patterns were mostly similar; people in both regions purchased the same foods frequently and common staple foods and tea with sugar provided most of the energy and nutrients.

The significant differences in micronutrient intake may be due to cultural dietary habits or seasonality in the two regions. In Vihiga, people consumed significantly more vitamins $B_{12}$ and C and had a lower risk of insufficiency of these nutrients than in Kitui. Higher consumption of fish and other animal-source foods (ASFs) seemed to influence the higher vitamin $B_{12}$ intake. People in Vihiga habitually consumed small fish called *omena*. Moreover, the intake and frequency of ASFs such as meats, poultries, eggs, fish, and milk were higher in Vihiga. In Kitui, the variety of fruits and vegetables that contribute to vitamin C intake was limited and the frequency of vegetable consumption was low. Only four fruits and vegetables contributed to vitamin C intake in Kitui, while there were nine items in Vihiga. That may have influenced the lower intake of vitamin C. These regional characteristics were consistent with those of a previous study conducted in the same regions [15]. Thus, geographically, culturally, and seasonally, the Vihiga people had better access to nutritious foods than the Kitui people.

Total protein intake did not differ between the two regions with about 60 g per day; however, their sources differed partially. Since the survey period was the harvest season for pigeon peas, people in Kitui consumed it in various forms almost daily. These dishes provided 38% of protein intake. The contributions of ASF-containing dishes were relatively low in both regions but lower in Kitui (a total of 19% in Kitui and 36% in Vihiga). In addition, people in Vihiga drank milk tea more frequently, especially when snacking. This supplied 18% protein intake compared to 8% in Kitui. Whether this was a regional difference or coincidence is unknown; however, these characteristics influenced the differences in protein sources. The health effects of the different protein sources should be investigated in future studies with a large sample.

Meal frequency and daily energy intake were similar; however, mealtime differed slightly between the two groups. In Kitui, the variations in mealtimes were ambiguous, especially during the daytime. On the other hand, there were three apparent peaks of mealtimes in Vihiga; 8:00 (breakfast), 13:00–14:00 (lunch) and 19:00 (dinner). The energy intake at breakfast in Kitui tended to be lower than that in Vihiga. In addition, the energy intake of breakfast was lower than that of dinner in Kitui only ($p < 0.05$, Wilcoxon signed-rank test). The participants in Vihiga tended to eat energy-dense foods such as *mandazi* (African doughnut) and cassava for breakfast, which may have caused the higher energy intake. Some participants in Kitui managed the kiosk as well as farming, therefore there were large individual differences in lifestyles and irregular mealtimes. Recent studies have suggested the possible impact of meal timing on body weight change in Western countries [20,21]. There may be room for examining this relationship in rural Kenya to prevent obesity in the future.

Both regions had a high risk of insufficient calcium and vitamin A intake. Some foods or dishes that contributed to the intake of these nutrients were somewhat self-sufficient, although seasonality may be a factor; pigeon pea stew, egg, kale, papaya and milk tea for Kitui; a variety of green leaves, egg, mango, and milk tea for Vihiga. The present study showed that most fruits consumed were homegrown but the consumption itself was low. Such foods would suit each region's natural characteristics and eating habits; therefore, increasing the home production and consumption of these foods would contribute to improved diets. Improving egg and milk productivity may be an essential approach. A study that analyzed the nutritional profile of eight African leafy vegetables showed that

they could considerably contribute to the requirements of vitamin A and iron [8]. Many foods have potential other than those mentioned above; however, many were neglected and underutilized [6,7,22]. A review identified the following constraints to the use of these crops: lack of awareness, low domestication, and low priority in the food system [6]. Gewa et al. reported that the lack of availability and poor taste of indigenous and traditional foods were the reason for the low intake [23]. It was also reported that adults viewed many indigenous fruits as food for children [24]. There is a need to reaffirm every local product, communicate about their availability and use from generation to generation and develop recipes.

The variety of dishes was greater in Vihiga. However, both regions had similar dietary patterns. Their typical breakfast consisted of milk tea with bread or *mandazi*. At lunch and dinner, *ugali*, rice, and *chapati* were common staple foods and mainly accompanied by side dishes, such as beef stew, bean stew, or stir-fried vegetables (mostly kale). In both regions, *ugali*, *muthokoi* (only in Kitui) and milk tea with sugar accounted for about 40% of energy and protein intakes and their portion sizes were also large at more than 300 g. The results revealed that their diet was dependent on a high-energy density diet, with a limited number of ingredients, dishes, and cooking methods. That may contribute to the high prevalence of obesity [25]. Keding et al. emphasized that high sugar consumption among Kenyans is likely responsible for the increase in being overweight or obese [10,26]. Reducing such energy-density foods will be recommended to address both obesity and micronutrient deficiencies other than promoting nutrient-rich foods.

The consumption frequency did not necessarily correlate with the frequency of home production. Ng'endo et al. also reported a more significant impact on food consumption by wealth status, ethnicity and education level than by food production [12]. Tomato, onion (they used tomato and onion as relishes), rice, cabbage, and Irish potatoes were commonly consumed but often purchased in the two regions. A previous study observed a similar trend: in rural Tanzania, Keding et al. found that the link between production and consumption was only for cultivated traditional vegetables but not for exotic vegetables such as tomatoes and onions [27]. Moreover, they purchased and consumed sugar with tea almost every day. This may be because both regions have high affordability, usability, and preference for these foods. The study participants seemed to spend money on the same food every day. Consuming homegrown products could allow them to spend more money on other nutrient-rich foods such as ASFs. Maximizing the use of local products and combining them with other foods can lead to a sustainable supply of adequate nutrients.

The survey has participant observation, which is a strength of this study. Dietary intake was recorded as accurately as possible. The 24 h recalls, commonly used to assess dietary intake in rural areas, were reported to both under- and overestimate compared to the weighed-food record [28]. This study observed the weights of ingredients and dishes and cooking flows. All information, including the situation, was recorded. We believe that this study is valuable in providing information on what, when, and how much was eaten, from the nutrient to the dish level.

This study has several limitations. First, the sample size was quite small and selective; therefore, the study results cannot be interpreted as a simple comparison of two rural populations. The households were selected because of their willingness to participate in research, accessibility and safety, and the women preparing meals were the subjects. The prevalence of being overweight or obese among study participants was higher than reported in a national survey [2]. Additionally, we could not determine the economic status of the participants, which may have affected their dietary intake. The present study might have included only households that could afford to provide food and livelihood for the enumerators, although the participants received an allowance of two USD per day for the living expenses of the enumerators at the end of the survey period. However, as the previous study's dietary characteristics in the same regions were similar [13], the target households were considered close to the region's general population. Second, since the enumerators stayed at the targeted households during the survey, there is no denying the possibility of its influence on dietary intake. To avoid this, we asked them to lead as normal

a life as possible. Third, the study was conducted only during one season and did not provide an overall picture of dietary habits or the degree of use of local foods throughout the year, so caution should be required in its interpretation. Our survey conducted in March, during the rainy season, showed slightly different fruit types and intakes. Gewa et al. reported the seasonal differences in food security and production of maize, beans and vegetables in other rural areas of Kenya [29]. In fact, in Kitui, the survey period was during the harvest season of pigeon peas, which strongly influenced dietary intake. In contrast, Ng'endo reported no seasonal differences in food procurement and home consumption in western Kenya [12]. The results of other seasonal surveys may differ, especially in Kitui. Fourth, the nutritional fortification of maize and cooking oil was not considered. This study showed participants reported to have used own-produced maize (Vihiga: 70% and Kitui: 38% of the total use). However, we calculated nutrient intake with unfortified maize since we could not clearly determine whether they consumed fortified maize. Therefore, the micronutrient intake may have been underestimated, especially in Kitui, where the proportion of own-produced maize was low. The frequency of own-produced maize consumption may differ depending on the season; therefore, it is essential to confirm the use of fortified products by season in the future. Finally, information on food sources was self-reported; therefore, some recall bias might exist.

## 5. Conclusions

We clarified the influences of local resource use and dietary patterns on regional differences in dietary intake. In Vihiga, the habits of consuming small fish and production and home consumption of a greater variety of African leafy vegetables were considered to influence their higher micronutrient intake. The people in Kitui consumed large quantities of harvested peas, contributing to energy and micronutrient intake. Fruits were primarily home-produced but their consumption was not so high in both regions. Nevertheless, dietary patterns were similar; common staple dishes and tea with sugar accounted for about 40% of energy and protein intakes. There was no difference in the foods purchased frequently. Reaffirmation of available local resources and increased production and consumption of local fruits and vegetables would contribute to a sustainable supply of adequate micronutrients. Further studies are required to develop strategies to promote healthy dietary habits and improve health status.

**Author Contributions:** Conceptualization, A.H., K.H., K.I., P.M. and Y.M.; methodology, A.H., K.H., K.I. and Y.M.; formal analysis, M.K. and M.H.; investigation, M.K., M.H., A.H. and Y.M.; writing—original draft preparation, M.K. and A.H.; writing—review and editing, Y.T., K.I.-T., P.M. and Y.M.; supervision, Y.M.; project administration, A.H., K.H., K.I., P.M. and Y.M.; funding acquisition, Y.M. All authors have read and agreed to the published version of the manuscript.

**Funding:** This study was supported by the Japan Ministry of Agriculture, Forestry, and Fisheries. (Grant number: L21ROM109) "Stimulating use of local resources in Africa to improve nutrition and livelihoods: A new integrated food consumption assessment tool for better decision making in nutrition interventions".

**Institutional Review Board Statement:** The study was conducted in accordance with the Declaration of Helsinki and approved by the Ethics Committee of the Tokyo University of Agriculture (Approval No. 1825) and the KNH-UoN Ethics and Research Committee (Approval No. KNH-ERC/A/129).

**Informed Consent Statement:** Informed consent was obtained from all subjects involved in the study.

**Data Availability Statement:** The data presented in this study are available on reasonable request from the corresponding author.

**Acknowledgments:** The authors gratefully acknowledge all the participants for their cooperation.

**Conflicts of Interest:** The authors declare no conflict of interest.

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
