# Peer review of "Characteristics of Dietary Intake in Relation to the Consumption of Home-Produced Foods among Farm Women in Two Rural Areas of Kenya: A Preliminary Study"

_2674-0311, doi:10.3390/dietetics1030021_

Round 1

Reviewer 1 Report

Review

The authors attempted to estimate the share of food produced at home in relation to food consumed, which is valuable. On the other hand, the issue in the article was treated quite generally both in the description of the results and in the discussion. The conclusion is hindered by the small study group, which indicates that this may be preliminary research. Already in the title of the paper this should have been informed.

Below are more specific comments that need to be addressed to make the content clearer.

Study design

Implementation of the survey in the dry season does not fully achieve the objectives of the survey. For example, it is difficult to estimate "the degree of home production of each ingredient." The authors themselves write: “Since food is scarce, people might purchase food to some extent”. Later on in the discussion, it is worth deepening this topic, for example, citing data informing about the scale of home-supply in the country and relating the results of the own study to this.

Section 2.2.

It would be good if the characteristics of food consumption in the two regions were compared, that is, which products occurred in both regions, and only then should there be a description of the differences. This will make it easier to see how food consumption differs between the two ethnic group

Section 2.4.

I wonder why the authors have written about measuring the health status of the members of the targeted households, when this information does not appear in the article.

Section 2.5.

To be honest, I don't know how women's food consumption was measured. The authors have mentioned about measuring for the whole household. Please give a more precise description.

Section 2.6.

How the degree of home production of each ingredient was measured?

Figure 2 and 3 – Why the home production of each food item is presented in the graph as a frequency. It seems that it would be better to show what % of the food consumed comes from home production

Discussion

The first sentence in the discussion - this sounds more like a hypothesis and not a result. In addition, there were no differences in energy intake after taking into account the two regions.

“Food consumption was not necessarily related  to food production”.  This is a very general statement. What is needed are details - what kind of food is being produced.

The phrase "both regions purchased the same foods frequently" needs improvement. It was the people who bought the food.

It is worth explaining why “In Kitui, the variety of fruits and vegetables that contribute to vitamin C intake was limited”

Please explain why “culturally the Vihiga people had better access  to nutritious foods than the Kitui people”.

The conclusions are too general.  They need to be improved, it is necessary to point out the cultural differences between ethnic groups, the possibilities of production due to geographical conditions , and then indicate what food should be produced and consumed to reduce the risk of deficiencies and, above all, reduce overweight and obesity.

Reviewer 2 Report

REVIEW

dietetics-1884968-peer-review-v1

The paper’s research topic is relevant to public health nutrition. However, some important information is lacking, and some other changes are necessary.

Random sampling is not described. I assume it was done as the Authors report probabilities, so it is obligatory. Otherwise, the probabilities and the indicated tests must be removed.

How the enumerators separate the food prepared for the household and the food eaten by the selected women is not clear.

The sample size is very limited, but the present work could provide much more information on the cooking methods. These can influence the retention of nutrients and affect the final nutrient content. Can the Authors elaborate on this?

The limitation on seasonal conditions is acknowledged, but it would be opportune for collecting other information on the usual use of the different food groups. Did the Authors this work? If yes, please add the appropriate sentences.

Line 366 “Fourth, the presence of nutritional fortification was not considered.” Authors say. Conversely, in lines 138-142, it is written, “We calculated the daily energy and nutrient intake per person using the Kenya Food 138 Composition Tables [13]. Kenya has commercially available nutritional fortified products such as maize and wheat flour, salt, oil, and fat [14]. In this study, we carried out nutritional calculations with fortified wheat flour, iodized salt, fortified margarine, unfortified maize flour, and cooking oil.” Moreover, in table 5, the footnote “calculated with fortified product” concerns Margarine on the bread. Please, clarify.

Finally, Introduction and Discussion can be enriched by considering the following papers concerning the healthy adult general population group in Kenya:

Ebere et al., 2017 http://repository.must.ac.ke/handle/123456789/245

Kigutha, 1997 https://academic.oup.com/ajcn/article/65/4/1168S/4655706?login=false

Keding, 2016 https://www.karger.com/Article/Abstract/442073#  

Ng’endo et al., 2016 https://doi.org/10.1080/03670244.2016.1200037

Muange and Ngigi, 2021 https://doi.org/10.1007/s12571-021-01174-8

Reviewer 3 Report

This study evaluated the differences in nutritional intake, influence of local products and dietary patterns between two rural Kenyan regions. The Kenyan population has seen a sharp increase in overweight individuals in recent decades, while malnutrition by default persists, especially in children. In this context considering local resources and dietary patterns is an essential aspect of nutritional assessment. The study is interesting, quite innovative and can be published, but first a few adjustments are necessary

1. There is an important, though not significant, difference between the BMIs of the two groups. At the same time, the calorie intake is higher among the Vihiga population. How can this inconsistency be explained? Is a different energy expenditure for work possible between the two groups of women?

2. The Kituiha population consumes more legumes while the Vihiga consume more fish. What are the health implications of these differences?

3. In the discussion, it should also be further explored and discussed what can be done to reduce the nutritional status of overweight individuals and reduce the risk of malnutrition. 

Round 2

Reviewer 1 Report

Dear Authors,

Thank you for taking my comments into consideration. 

Author Response

Dear Reviewer1,

Thank you for your good suggestion.

Reviewer 2 Report

I much appreciate the thorough revision done by the Authors. I have just a couple of suggestions.

The first one concerns the reporting of probability. When a random selection has not been done, the probability simply is not generated.

 Thus, it is incorrect to put it. In the specific case of the present paper, it is negligible to report the probability (the packages automatically report probability as they assume the sample is randomly selected) as it is evident that the intervals around the mean overlap (it is the reason for non-significance). In the few cases of significance, the intervals clearly do not overlap each other. I mean, the probability does not add any helpful information. Please remove it and comment on the data accordingly.

The second one suggests that all data are clearly indicated as 'responsible for food preparation' women's consumption. Otherwise, it can be confused with the general population's dietary pattern. Please, go through the methods and the results, including the tables.
